# Learning Implicitly Recurrent CNNs Through Parameter Sharing

**Pedro Savarese**
TTI-Chicago
savarese@ttic.edu

**Michael Maire**
University of Chicago
mmaire@uchicago.edu

## Abstract

We introduce a parameter sharing scheme, in which different layers of a convolutional neural network (CNN) are defined by a learned linear combination of parameter tensors from a global bank of templates. Restricting the number of templates yields a flexible hybridization of traditional CNNs and recurrent networks. Compared to traditional CNNs, we demonstrate substantial parameter savings on standard image classification tasks, while maintaining accuracy.

Our simple parameter sharing scheme, though defined via soft weights, in practice often yields trained networks with near strict recurrent structure; with negligible side effects, they convert into networks with actual loops. Training these networks thus implicitly involves discovery of suitable recurrent architectures. Though considering only the design aspect of recurrent links, our trained networks achieve accuracy competitive with those built using state-of-the-art neural architecture search (NAS) procedures.

Our hybridization of recurrent and convolutional networks may also represent a beneficial architectural bias. Specifically, on synthetic tasks which are algorithmic in nature, our hybrid networks both train faster and extrapolate better to test examples outside the span of the training set.

## 1 Introduction

The architectural details of convolutional neural networks (CNNs) have undergone rapid exploration and improvement via both human hand-design (Simonyan & Zisserman, 2015; Szegedy et al., 2015; He et al., 2016; Huang et al., 2017; Zhu et al., 2018) and automated search methods (Zoph & Le, 2017; Liu et al., 2018). Yet, this vast array of work limits itself to a circuit-like view of neural networks. Here, a CNN is regarded as a fixed-depth feed-forward circuit, with a distinct parameter governing each internal connection. These circuits are often trained to perform tasks which, in a prior era, might have been (less accurately) accomplished by running a traditional computer program coded by humans. Programs, and even traditional hardware circuits, have a more reusable internal structure, including subroutines or modules, loops, and associated control flow mechanisms.

We bring one aspect of such modularity into CNNs, by making it possible to learn a set of parameters that is reused across multiple layers at different depths. As the pattern of reuse is itself learned, our scheme effectively permits learning the length (iteration count) and content of multiple loops defining the resulting CNN. We view this approach as a first step towards learning neural networks with internal organization reminiscent of computer programs. Though we focus solely on loop-like structures, leaving subroutines and dynamic control flow to future work, this simple change suffices to yield substantial quantitative and qualitative benefits over the standard baseline CNN models.

While recurrent neural networks (RNNs) possess a loop-like structure by definition, their loop structure is fixed a priori, rather than learned as part of training. This can actually be a disadvantage in the event that the length of the loop is mismatched to the target task. Our parameter sharing scheme for CNNs permits a mix of loops and feed-forward layers to emerge. For example, trained with our scheme, a 50-layer CNN might learn a 2-layer loop that executes 5 times between layers 10 and 20, a 3-layer loop that runs 4 times from layers 30 to 42, while leaving the remaining layers to assume independent parameter sets. Our approach generalizes both CNNs and RNNs, creating a hybrid.

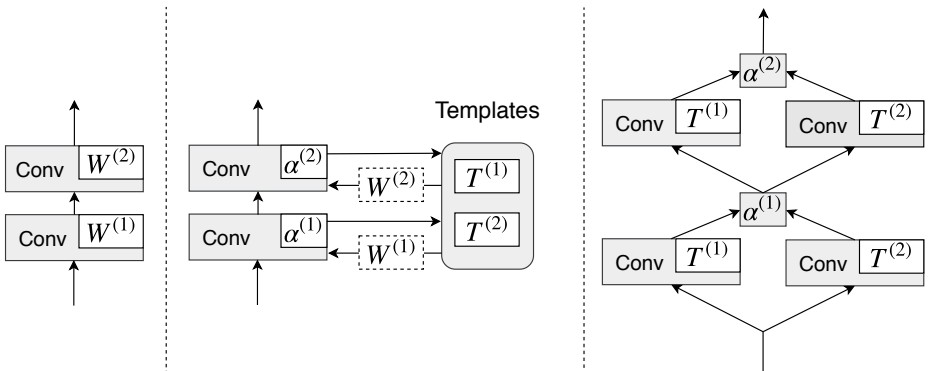

Figure 1: Parameter sharing scheme. **Left:** A CNN (possibly a variant such as a residual network), with each convolutional layer $i$ containing an individual parameter set $\mathbf{W}^{(i)}$. **Middle:** Parameter sharing among layers, where parameter templates $\mathbf{T}^{(1)}, \mathbf{T}^{(2)}$ are shared among each layer $i$, which now only contains a 2-dimensional parameter $\boldsymbol{\alpha}^{(i)}$. Weights $\mathbf{W}^{(i)}$ (no longer parameters, illustrated with dotted boxes) used by layer $i$ are generated from $\boldsymbol{\alpha}^{(i)}$ and templates $\mathbf{T}^{(1)}, \mathbf{T}^{(2)}$. **Right:** If weights $\mathbf{W}^{(i)}$ are outputs of a linear function (as in our method), learning parameter templates can be viewed as learning layer templates, offering a new (although equivalent) perspective for the middle diagram. Non-linearities are omitted for simplicity.

Figure 1 diagrams the parameter sharing scheme facilitating this hybridization. Inspired by dictionary learning, different network layers share, via weighted combination, global parameter templates. This re-parameterization is fully differentiable, allowing learning of sharing weights and template parameters. Section 3 elaborates, and also introduces tools for analyzing learned loop structures.

Section 4 demonstrates advantages of our hybrid CNNs across multiple experimental settings. Taking a modern CNN design as a baseline, and re-parameterizing it according to our scheme improves:

- **Parameter efficiency.** Here, we experiment with the standard task of image classification using modern residual networks (He et al., 2016; Zagoruyko & Komodakis, 2016). This task is a good proxy for general usefulness in computer vision, as high-performance classification architectures often serve as a backbone for many other vision tasks, such as semantic segmentation (Chen et al., 2016; Zhao et al., 2017).

  Our parameter sharing scheme drastically reduces the number of unique parameters required to achieve a given accuracy on CIFAR (Krizhevsky, 2009) or ImageNet (Russakovsky et al., 2015) classification tasks. Re-parameterizing a standard residual network with our scheme cuts parameters, without triggering any drop in accuracy. This suggests that standard CNNs may be overparameterized in part because, by design (and unlike RNNs), they lack capacity to learn reusable internal operations.

- **Extrapolation and generalization.** Here, we explore whether our hybrid networks expand the class of tasks that one can expect to train neural networks to accomplish. This line of inquiry, focusing on synthetic tasks, shares motivations with work on Neural Turing Machines (Graves et al., 2014). Specifically, we would like neural networks to be capable of learning to perform tasks for which there are concise traditional solution algorithms. Graves et al. (2014) uses sorting as an example task. As we examine an extension of CNNs, our tasks take the form of queries about planar graphs encoded as image input.

  On these tasks, we observe improvements to both generalization ability and learning speed for our hybrid CNNs, in comparison to standard CNNs or RNNs. Our parameter sharing scheme, by virtue of providing an architectural bias towards networks with loops, appears to assist in learning to emulate traditional algorithms.

An additional side effect, seen in practice in many of our experiments, is that two different learned layers often snap to the same parameter values. That is, layers $i$ and $j$, learn coefficient vectors $\boldsymbol{\alpha}^{(i)}$ and $\boldsymbol{\alpha}^{(j)}$ (see Figure 1) that converge to be the same (up to scaling). This is a form of architecture discovery, as it permits representation of the CNN as a loopy wiring diagram between repeated layers. Section 4.3 presents example results. We also draw comparisons to existing neural architec-

ture search (NAS) techniques. By simply learning recurrent structure as byproduct of training with standard stochastic gradient descent, we achieve accuracy competitive with current NAS procedures.

Before delving into the details of our method, Section 2 provides additional context in terms of prior work on recurrent models, parameter reduction techniques, and program emulation. Sections 3 and 4 describe our hybrid shared-parameter CNN, experimental setup, and results. Section 5 concludes with commentary on our results and possible future research pathways.[1]

## 2  RELATED WORK

Recurrent variants of CNNs are used extensively for visual tasks. Recently, Zamir et al. (2017) propose utilizing a convolutional LSTM (Shi et al., 2015) as a generic feedback architecture. RNN and CNN combinations have been used for scene labeling (Pinheiro & Collobert, 2014), image captioning with attention (Xu et al., 2015), and understanding video (Donahue et al., 2015), among others. These works combine CNNs and RNNs at a coarse scale, and in a fixed hand-crafted manner. In contrast, we learn the recurrence structure itself, blending it into the inner workings of a CNN.

Analysis of residual networks (He et al., 2016) reveals possible connections to recurrent networks stemming from their design (Liao & Poggio, 2016). Greff et al. (2017) provide evidence that residual networks learn to iteratively refine feature representations, making an analogy between a very deep residual network and an unrolled loop. Jastrzebski et al. (2018) further explore this connection, and experiment with training residual networks in which some layers are forced to share identical parameters. This hard parameter sharing scheme again builds a predetermined recurrence structure into the network. It yields successfully trained networks, but does not exhibit the type of performance gains that Section 4 demonstrates for our soft parameter sharing scheme.

Closely related to our approach is the idea of hypernetworks (Ha et al., 2016), in which one part of a neural network is parameterized by another neural network. Our shared template-based re-parameterization could be viewed as one simple choice of hypernetwork implementation. Perhaps surprisingly, this class of ideas has not been well explored for the purpose of reducing the size of neural networks. Rather, prior work has achieved parameter reduction through explicit representation bottlenecks (Iandola et al., 2016), sparsifying connection structure (Prabhu et al., 2018; Huang et al., 2018; Zhu et al., 2018), and pruning trained networks (Han et al., 2016).

Orthogonal to the question of efficiency, there is substantial interest in extending neural networks to tackle new kinds of tasks, including emulation of computer programs. Some approach this problem using additional supervision in the form of execution traces (Reed & de Freitas, 2016; Cai et al., 2017), while other focus on development of network architectures that can learn from input-output pairs alone (Graves et al., 2014; 2016; Zaremba et al., 2016; Trask et al., 2018). Our experiments on synthetic tasks fall into the latter camp. At the level of architectural strategy, Trask et al. (2018) benefit from changing the form of activation function to bias the network towards correctly extrapolating common mathematical formulae. We build in a different implicit bias, towards learning iterative procedures within a CNN, and obtain a boost on correctly emulating programs.

## 3  SOFT PARAMETER SHARING

In convolutional neural networks (CNNs) and variants such as residual CNNs (ResNets) (He et al., 2016) and DenseNets (Huang et al., 2017), each convolutional layer $i$ contains a set of parameters $\mathbf{W}^{(i)}$, with no explicit relation between parameter sets of different layers. Conversely, a strict structure is imposed to layers of recurrent neural networks (RNNs), where, in standard models (Hochreiter & Schmidhuber, 1997), a single parameter set $\mathbf{W}$ is shared among all time steps. This leads to a program-like computational flow, where RNNs can be seen as loops with fixed length and content. While some RNN variants (Graves et al., 2013; Koutník et al., 2014; Yang et al., 2018) are less strict on the length or content of loops, these are still typically fixed beforehand.

As an alternative to learning hard parameter sharing schemes – which correspond to the strict structure present in RNNs – our method consists of learning soft sharing schemes through a relaxation of

---

[1]Our code is available at https://github.com/lolemacs/soft-sharing

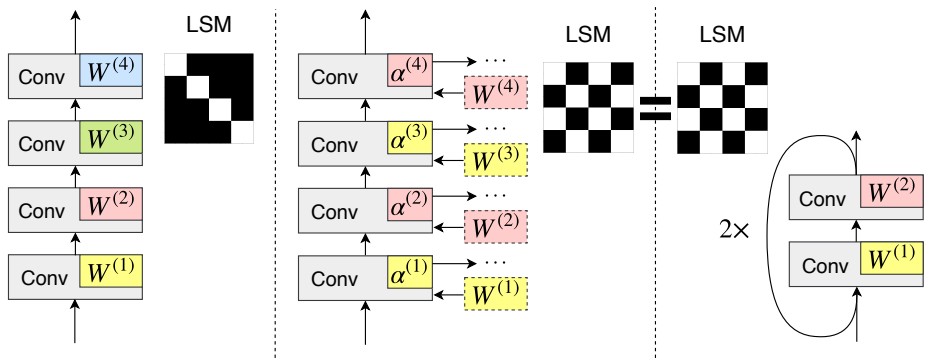

Figure 2: Connection between the LSM matrix $S$ ( where $S_{i,j} = \frac{|\langle \boldsymbol{\alpha}^{(i)}, \boldsymbol{\alpha}^{(j)} \rangle|}{\|\boldsymbol{\alpha}^{(i)}\| \|\boldsymbol{\alpha}^{(j)}\|}$ ) and the structure of the network. White and black entries correspond to maximum and minimum similarities ($S_{i,j} = 1$ and $S_{i,j} = 0$, respectively). **Left:** Empirically, CNNs present no similarity between parameters of different layers. **Middle:** Trained with our method, the layer similarity matrix (LSM) captures similarities between different layers, including pairs with close to maximum similarity. Such pairs (depicted by same-colored coefficients and weights, and by white entries in the LSM) perform similar operations on their inputs. **Right:** We can tie together parameters of similar layers, creating a hard parameter sharing scheme. The network can then be folded, creating self-loops and revealing an explicit recurrent computation structure.

this structure. We accomplish this by expressing each layer's parameters $\mathbf{W}^{(i)}$ as a linear combination of parameter templates $\mathbf{T}^{(1)}, \ldots, \mathbf{T}^{(k)}$, each with the same dimensionality as $\mathbf{W}^{(i)}$:

$$\mathbf{W}^{(i)} := \sum_{j=1}^{k} \alpha_j^{(i)} \mathbf{T}^{(j)} \tag{1}$$

where $k$ is the number of parameter templates (chosen freely as a hyperparameter) and $\boldsymbol{\alpha}^{(i)}$, a $k$-dimensional vector, is the coefficients of layer $i$. Figure 1 (left and middle) illustrates the difference between networks trained with and without our method. This relaxation allows for coefficients and parameter templates to be (jointly) optimized with gradient-based methods, yielding negligible extra computational cost, with a single constraint that only layers with same parameter sizes can share templates. Note that constraining coefficients $\boldsymbol{\alpha}^{(i)}$ to be one-hot vectors leads to hard sharing schemes, at the cost of non-differentiability.

Having $k$ as a free parameter decouples the number of parameters in network from its depth. Typically, $L$ convolutional layers with constant channel and kernel sizes $C, K$ have $O(LC^2K^2)$ total parameters. Our soft sharing scheme changes the total number of parameters to $O(kL + kC^2K^2) = O(kC^2K^2)$. Sections 4.1 and 4.2 show that we can decrease the parameter count of standard models without significantly impacting accuracy, or simply attain higher accuracy with $k = L$.

In the next two subsections, we discuss two consequences of the linearity of Equation (1). First, it enables alternative interpretations of our method. Second, and a major advantage, as is the case in many linear relaxations of integer problems, we are able to extract hard sharing schemes in practice, and consequently detect implicit self-loops in a CNN trained with our method.

### 3.1 INTERPRETATION

For layers $i$ that are linear in $\mathbf{W}^{(i)}$ (*e.g.* matrix multiplication, convolution), we can view our method as learning template layers which are shared among a network. More specifically, for a convolutional layer $\mathbf{U}^{(i)}(\mathbf{X}) = \mathbf{W}^{(i)} * \mathbf{X}$, and considering Equation (1):

$$\mathbf{U}^{(i)}(\mathbf{X}) = \mathbf{W}^{(i)} * \mathbf{X} = \sum_{j=1}^{k} \alpha_j^{(i)} \mathbf{T}^{(j)} * \mathbf{X} \tag{2}$$

where $\mathbf{T}^{(j)} * \mathbf{X}$, the result of a convolution with filter sets $\mathbf{T}^{(j)}$, can be seen as the output of a template layer with individual parameters $\mathbf{T}^{(j)}$. Such layers can be seen as global feature extractors,

and coefficients $\boldsymbol{\alpha}^{(i)}$ determine which features are relevant for the $i$'th computation of a network. This is illustrated in Figure 1 (right diagram).

This view gives a clear connection between coefficients $\boldsymbol{\alpha}$ and the network's structure. Having $\boldsymbol{\alpha}^{(i)} = \boldsymbol{\alpha}^{(i+2)}$ yields $\mathbf{W}^{(i)} = \sum_{j=1}^{k} \alpha_j^{(i)} \mathbf{T}^{(j)} = \sum_{j=1}^{k} \alpha_j^{(i+2)} \mathbf{T}^{(j)} = \mathbf{W}^{(i+2)}$, and hence layers $i$ and $i + 2$ are functionally equivalent. Such a network can be folded to generate an equivalent model with two layers and a self-loop, an explicitly recurrent network. While this is also possible for networks without parameter sharing, a learned alignment of $C^2 K^2$ parameters is required (unlikely in practice), instead of aligning only $k \leq L$ parameters.

## 3.2 IMPLICIT RECURRENCES

To identify which layers in a network perform approximately the same operation, we can simply check whether their coefficients are similar. We can condense this information for all pairs of layers $i, j$ in a similarity matrix $S$, where $S_{i,j} = s(\boldsymbol{\alpha}^{(i)}, \boldsymbol{\alpha}^{(j)})$ for some similarity measure $s$.

For networks with normalization layers, the network's output is invariant to weight rescaling. In this setting, a natural measure is $s(\boldsymbol{\alpha}^{(i)}, \boldsymbol{\alpha}^{(j)}) = \frac{|\langle \boldsymbol{\alpha}^{(i)}, \boldsymbol{\alpha}^{(j)} \rangle|}{\|\boldsymbol{\alpha}^{(i)}\| \|\boldsymbol{\alpha}^{(j)}\|}$ (absolute value of cosine similarity), since it possess this same property.[2] We call $S$ the layer similarity matrix (LSM). Figure 2 illustrates and Section 4.3 shows experimentally how it can be used to extract recurrent loops from trained CNNs.

While structure might emerge naturally, having a bias towards more structured (recurrent) models might be desirable. In this case, we can add a *recurrence regularizer* to the training objective, pushing parameters to values which result in more structure. For example, we can add the negative of sum of elements of the LSM: $\mathcal{L}_R = \mathcal{L} - \lambda_R \sum_{i,j} S_{i,j}$, where $\mathcal{L}$ is the original objective. The larger $\lambda_R$ is, the closer the elements of $S$ will be to 1. At an extreme case, this regularizer will push all elements in $S$ to 1, resulting in a network with a single layer and a self-loop.

## 4 EXPERIMENTS

We begin by training variants of standard models with soft parameter sharing, observing that it can offer parameter savings with little impact on performance, or increase performance at the same parameter count. Section 4.3 demonstrates conversion of a trained model into explicitly recurrent form. We then examine synthetic tasks (Section 4.4), where parameter sharing improves generalization. Appendix B contains details on the initialization for the coefficients $\boldsymbol{\alpha}$.

### 4.1 CLASSIFICATION ON CIFAR

The CIFAR-10 and CIFAR-100 datasets (Krizhevsky, 2009) are composed of $60,000$ colored $32 \times 32$ images, labeled among 10 and 100 classes respectively, and split into $50,000$ and $10,000$ examples for training and testing. We pre-process the training set with channel-wise normalization, and use horizontal flips and random crops for data augmentation, following He et al. (2016).

Using Wide ResNets (WRN) (Zagoruyko & Komodakis, 2016) as a base model, we train networks with the proposed soft parameter sharing method. Since convolution layers have different number of channels and kernel sizes throughout the network, we create 3 layer groups and only share templates among layers in the same group. More specifically, WRNs for CIFAR consist of 3 stages whose inputs and outputs mostly have a constant number of channels ($C$, $2C$ and $4C$, for some $C$). Each stage contains $\frac{L-4}{3}$ layers for a network with depth $L$, hence we group layers in the same stage together, except for the first two, a residual block whose input has a different number of channels.

Thus, all layers except for the first 2 in each stage perform parameter sharing (illustrated in left diagram of Figure 4). Having $k$ templates per group means that $\frac{L-4}{3} - 2$ convolution layers share $k$ parameter templates. We denote by SWRN-$L$-$w$-$k$ a WRN with $L$ layers, widen factor $w$ and $k$ parameter templates per group (trained with our method). Setting $k = \frac{L-4}{3} - 2$ means we have

---

[2]We take the absolute value for simplicity: while negating a layer's weights can indeed impact the network's output, this is circumvented by adding a $-1$ multiplier to, for example, the input of layer $i$ in case $\langle \boldsymbol{\alpha}^{(i)}, \boldsymbol{\alpha}^{(j)} \rangle$ is negative, along with $\boldsymbol{\alpha}^{(i)} \leftarrow -\boldsymbol{\alpha}^{(i)}$.

Table 1: Test error (%) on CIFAR-10 and CIFAR-100. SWRN 28-10, the result of training a WRN 28-10 with our method and one template per layer, significantly outperforms the base model, suggesting that our method aids optimization (both models have the same capacity). SWRN 28-10-1, with a single template per sharing group, performs close to WRN 28-10 while having significantly less parameters and capacity. * indicates models trained with dropout $p = 0.3$ (Srivastava et al., 2014). Results are average of 5 runs.

| CIFAR | Params | C-10+ | C-100+ |
|---|---|---|---|
| WRN 28-10 | 36M | 4.0 | 19.25 |
| WRN 28-10* | 36M | 3.89 | 18.85 |
| SWRN 28-10 | 36M | **3.74** | 18.78 |
| SWRN 28-10* | 36M | 3.88 | **18.43** |
| SWRN 28-10-1 | 12M | 4.01 | 19.73 |

Table 2: Performance of wider SWRNs. Parameter reduction ($k = 2$) leads to lower errors for CIFAR-10, with models being competitive against newer model families that have bottleneck layers, group convolutions, or many layers. Best SWRN results are in bold, and best overall results are underlined.

| CIFAR | Params | C-10+ | C-100+ |
|---|---|---|---|
| ResNeXt-29 16x64 | 68M | 3.58 | 17.31 |
| DenseNet 100-24 | 27M | 3.74 | 19.25 |
| DenseNet 190-40 | 26M | 3.46 | 17.18 |
| SWRN 28-10* | 36M | 3.88 | 18.43 |
| SWRN 28-10-2* | 17M | 3.75 | 18.66 |
| SWRN 28-14* | 71M | 3.67 | 18.25 |
| SWRN 28-14-2* | 33M | 3.69 | 18.37 |
| SWRN 28-18* | 118M | 3.48 | **17.43** |
| SWRN 28-18-2* | 55M | **3.43** | 17.75 |

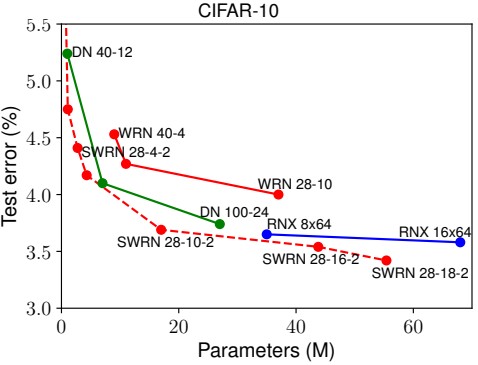 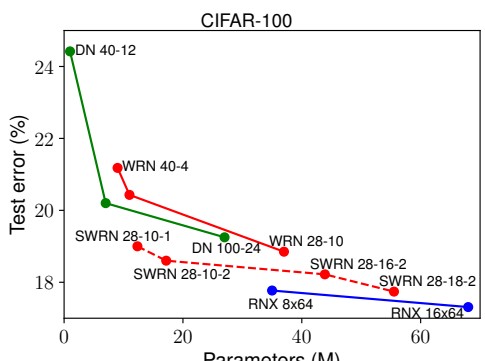

Figure 3: Parameter efficiency for different models. On both CIFAR-10 and CIFAR-100, SWRNs are significantly more efficient than WRNs. DN and RNX denotes DenseNet and ResNeXt, respectively, and are plotted for illustration: both models employ orthogonal efficiency techniques, such as bottleneck layers. Best viewed in color.

one parameter template per layer, and hence no parameter reduction. We denote SWRN-$L$-$w$ (thus omitting $k$) as a model in this setting.

Following Zagoruyko & Komodakis (2016), we train each model for 200 epochs with SGD and Nesterov momentum of 0.9 and a batch size of 128. The learning rate is initially set to 0.1 and decays by a factor of 5 at epochs 60, 120 and 160. We also apply weight decay of $5 \times 10^{-4}$ on all parameters except for the coefficients $\boldsymbol{\alpha}$.

Tables 1 and 2 present results. Networks trained with our method yield superior performance in the setting with no parameter reduction: SWRN 28-10 presents $6.5\%$ and $2.5\%$ lower relative test errors on C-10 and C-100, compared to the base WRN 28-10 model. With fewer templates than layers, SWRN 28-10-1 (all 6 layers of each group perform the same operation), performs virtually the same as the base WRN 28-10 network, while having $\frac{1}{3}$ of its parameters. On CIFAR-10, parameter reduction ($k = 2$) is beneficial to test performance: the best performance is achieved by SWRN 28-18-2 (3.43% test error), outperforming the ResNeXt-29 16x64 model (Xie et al., 2017), while having fewer parameters (55M against 68M) and no bottleneck layers.

Figure 3 shows that our parameter sharing scheme uniformly improves accuracy-parameter efficiency; compare the WRN model family (solid red) to our SWRN models (dotted red).

Table 4 presents a comparison between our method and neural architecture search (NAS) techniques (Zoph & Le, 2017; Xie et al., 2019; Liu et al., 2019; Pham et al., 2018; Real et al., 2018) on CIFAR-10 – results differ from Table 2 solely due to cutout (DeVries & Taylor, 2017), which is commonly used in NAS literature; NAS results are quoted from their respective papers. Our method outperforms architectures discovered by recent NAS algorithms, such as DARTS (Liu et al., 2019),

Table 3: *(below)* ImageNet classification results: training WRN 50-2 with soft parameter sharing leads to better performance by itself, without any tuning on the number of templates $k$. Top-1 and Top-5 errors (%) are computed using a single crop.

| ImageNet | Params | Top-1 | Top-5 |
|---|---|---|---|
| WRN 50-2 | 69M | 22.0 | 6.05 |
| DenseNet-264 | 33M | 22.15 | 6.12 |
| ResNet-200 | 65M | 21.66 | 5.79 |
| SWRN 50-2 | 69M | 21.74 | 5.95 |

Table 4: *(right)* Test error (%) on CIFAR-10 of SWRNs and models found via neural architecture search (NAS) (all trained with cutout). Networks trained with soft parameter sharing provide competitive performance against NAS methods while having low computational cost.

| CIFAR-10 | Params (M) | Training Time (GPU days) | Test Error (%) |
|---|---|---|---|
| NASNet-A | 3.3 | 1800 | 2.65 |
| NASNet-A | 27.6 | 1800 | 2.4 |
| AmoebaNet-B | 2.8 | 3150 | 2.55 |
| AmoebaNet-B | 13.7 | 3150 | 2.31 |
| AmoebaNet-B | 26.7 | 3150 | 2.21 |
| AmoebaNet-B | 34.9 | 3150 | 2.13 |
| DARTS | 3.4 | 4 | 2.83 |
| SNAS | 2.8 | 1.5 | 2.85 |
| ENAS | 4.6 | 0.45 | 2.89 |
| WRN 28-10 (baseline with cutout) | 36.4 | 0.4 | 3.08 |
| SWRN 28-4-2 | 2.7 | 0.12 | 3.45 |
| SWRN 28-6-2 | 6.1 | 0.25 | 3.0 |
| SWRN 28-10 | 36.4 | 0.4 | 2.7 |
| SWRN 28-10-2 | 17.1 | 0.4 | 2.69 |
| SWRN 28-14 | 71.4 | 0.7 | 2.55 |
| SWRN 28-14-2 | 33.5 | 0.7 | 2.53 |

SNAS (Xie et al., 2019) and ENAS (Pham et al., 2018), while having similarly low training cost. We achieve 2.69% test error after training less than 10 hours on a single NVIDIA GTX 1080 Ti. This accuracy is only bested by NAS techniques which are several orders of magnitude more expensive to train. Being based on Wide ResNets, our models do, admittedly, have more parameters.

Comparison to recent NAS algorithms, such as DARTS and SNAS, is particularly interesting as our method, though motivated differently, bears some notable similarities. Specifically, all three methods are gradient-based and use an extra set of parameters (architecture parameters in DARTS and SNAS) to perform some kind of soft selection (over operations/paths in DARTS/SNAS; over templates in our method). As Section 4.3 will show, our learned template coefficients $\alpha$ can often be used to transform our networks into an explicitly recurrent form - a discovered CNN-RNN hybrid.

To the extent that our method can be interpreted as a form of architecture search, it might be complementary to standard NAS methods. While NAS methods typically search over operations (*e.g.* activation functions; $3 \times 3$ or $5 \times 5$ convolutions; non-separable, separable, or grouped filters; dilation; pooling), our soft parameter sharing can be seen as a search over recurrent patterns (which layer processes the output at each step). These seem like orthogonal aspects of neural architectures, both of which may be worth examining in an expanded search space. When using SGD to drive architecture search, these aspects take on distinct forms at the implementation level: soft parameter sharing across layers (our method) vs hard parameter sharing across networks (recent NAS methods).

## 4.2 CLASSIFICATION ON IMAGENET

We use the ILSVRC 2012 dataset (Russakovsky et al., 2015) as a stronger test of our method. It is composed of 1.2M training and $50,000$ validation images, drawn from 1000 classes. We follow Gross & Wilber (2016), as in Zagoruyko & Komodakis (2016); Huang et al. (2017); Xie et al. (2017), and report Top-1 and Top-5 errors on the validation set using single $224 \times 224$ crops. For this experiment, we use WRN 50-2 as a base model, and train it with soft sharing and no parameter reduction. Having bottleneck blocks, this model presents less uniform number of channels of layer inputs and outputs. To apply our method, we group convolutions in 12 groups: for each of the 4 stages in a WRN 50-2, we create 3 groups, one for each type of layer in a bottleneck unit ($C \to B$, $B \to B$ and $B \to C$ channel mappings, for bottleneck $B$). Without any change in hyperparameters, the network trained with our method outperforms the base model and also deeper models such as DenseNets (though using more parameters), and performs close to ResNet-200, a model with four times the number of layers and a similar parameter count. See Table 3.

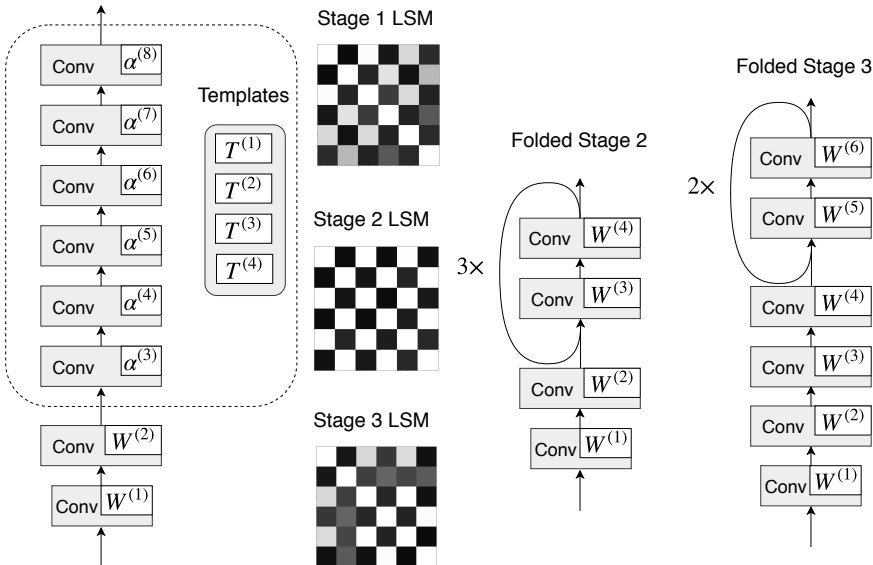

Figure 4: Extracting implicit recurrences from a SWRN 28-10-4. *Left:* Illustration of the stages of a SWRN-28-10-4 (residual connections omitted for clarity). The first two layers contain individual parameter sets, while the other six share four templates. All 3 stages of the network follow this structure. *Middle:* LSM for each stage after training on CIFAR-10, with many elements close to 1. Hard sharing schemes can be created for pairs with large similarity by tying their coefficients (or, equivalently, their effective weights). *Right:* Folding stages 2 and 3 leads to self-loops and a CNN with recurrent connections – LSM for stage 2 is a repetition of 2 rows/columns, and folding decreases the number of parameters.

### 4.3 LEARNING IMPLICIT RECURRENCES

Results on CIFAR suggest that training networks with few parameter templates $k$ in our soft sharing scheme results in performance comparable to the base models, which have significantly more parameters. The lower $k$ is, the larger we should expect the layer similarities to be: in the extreme case where $k = 1$, all layers in a sharing scheme have similarity 1, and can be folded into a single layer with a self-loop.

For the case $k > 1$, there is no trivial way to fold the network, as layer similarities depend on the learned coefficients. We can inspect the model's layer similarity matrix (LSM) and see if it presents implicit recurrences: a form of recurrence in the rows/columns of the LSM. Surprisingly, we observe that rich structures emerge naturally in networks trained with soft parameter sharing, *even without the recurrence regularizer*. Figure 4 shows the per-stage LSM for CIFAR-trained SWRN 28-10-4. Here, the six layers of its stage-2 block can be folded into a loop of two layers, leading to an error increase of only $0.02\%$. Appendix A contains an additional example of network folding, diversity of LSM patterns across different runs, and an epoch-wise evolution of the LSM, showing that many patterns are observable after as few as 5 epochs of training.

### 4.4 EVALUATION ON NATURALLY RECURRENT TASKS

While the propensity of our parameter sharing scheme to encourage learning of recurrent networks is a useful parameter reduction tool, we would also like to leverage it for qualitative advantages over standard CNNs. On tasks for which a natural recurrent algorithm exists, does training CNNs with soft parameter sharing lead to better extrapolation?

To answer this, we set up a synthetic algorithmic task: computing shortest paths. Examples are $32 \times 32$ grids containing two query points and randomly (with probability $0.1$) placed obstacles. The objective is to indicate which grid points belong to a shortest path between the query points.

We use curriculum learning for training, allowing us to observe how well each model adapts to more difficult examples as training phases progress. Moreover, for this task curriculum learning causes faster learning and superior performance for all trained models.

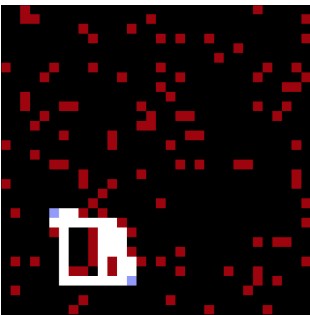

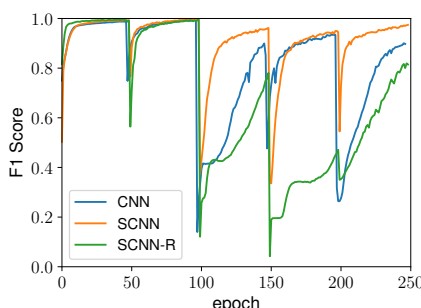

(a) Generated example for the synthetic shortest paths task. Blue pixels indicate the query points; red pixels represent obstacles, and white pixels are points in a shortest path (in terms of Manhattan distance) between query pixels. The task consists of predicting the white pixels (shortest paths) from the blue and red ones (queries and obstacles).

(b) Training curves for the shortest paths task, where difficulty of examples increases every 50 epochs. A SCNN adapts faster than a CNN to new phases and performs better, suggesting better extrapolation capacity. With a recurrence regularizer $\lambda_R = 0.01$ (SCNN-R), the model makes faster progress on the first phase, but fails to adapt to harder examples.

Figure 5: Shortest paths task. Best viewed in color.

Training consists of 5 curriculum phases, each one containing 5000 examples. The maximum allowed distance between the two query points increases at each phase, thus increasing difficulty. In the first phase, each query point is within a $5 \times 5$ grid around the other query point, and the grid size increases by 2 on each side at each phase, yielding a final grid size of $21 \times 21$ at phase 5.

We train a CNN, a CNN with soft parameter sharing and one template per layer (SCNN), and an SCNN with recurrence regularizer $\lambda_R = 0.01$. Each model trains for 50 epochs per phase with Adam (Kingma & Ba, 2015) and a fixed learning rate of $0.01$. As classes are heavily unbalanced and the balance itself changes during phases, we compare $F_1$ scores instead of classification error.

Each model starts with a $1 \times 1$ convolution, mapping the 2 input channels to 32 output channels. Next, there are 20 channel-preserving $3 \times 3$ convolutions, followed by a final $1 \times 1$ convolution that maps 32 channels to 1. Each of the 20 $3 \times 3$ convolutions is followed by batch normalization (Ioffe & Szegedy, 2015), a ReLU non-linearity (Nair & Hinton, 2010), and has a 1-skip connection.

Figure 5 shows one example from our generated dataset and the training curves for the 3 trained models: the SCNN not only outperforms the CNN, but adapts better to harder examples at new curriculum phases. The SCNN is also advantaged over a more RNN-like model: with the recurrence regularizer $\lambda_R = 0.01$, all entries in the LSM quickly converge 1, as in a RNN. This leads to faster learning during the first phase, but presents issues in adapting to difficulty changes in latter phases.

## 5 CONCLUSION

In this work, we take a step toward more modular and compact CNNs by extracting recurrences from feed-forward models where parameters are shared among layers. Experimentally, parameter sharing yields models with lower error on CIFAR and ImageNet, and can be used for parameter reduction by training in a regime with fewer parameter templates than layers. Moreover, we observe that parameter sharing often leads to different layers being functionally equivalent after training, enabling us to collapse them into recurrent blocks. Results on an algorithmic task suggest that our shared parameter structure beneficially biases extrapolation. We gain a more flexible form of behavior typically attributed to RNNs, as our networks adapt better to out-of-domain examples. Our form of architecture discovery is also competitive with neural architecture search (NAS) algorithms, while having a smaller training cost than state-of-the-art gradient-based NAS.

As the only requirement for our method is for a network to have groups of layers with matching parameter sizes, it can be applied to a plethora of CNN model families, making it a general technique with negligible computational cost. We hope to raise questions regarding the rigid definitions of

CNNs and RNNs, and increase interest in models that fall between these definitions. Adapting our method for models with non-uniform layer parameter sizes (Huang et al., 2017; Zhu et al., 2018) might be of particular future interest.

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

# Appendix

## A  ADDITIONAL RESULTS FOR IMPLICIT RECURRENCES

Section 4.3 presents an example of implicit recurrences and folding of a SWRN 28-10-4 trained on CIFAR-10, where, for example, the last 6 layers in the second stage of the network fold into 2 layers with a self-loop.

Figure 6 presents an additional example, where non-trivial recurrences (unlike the one in Figure 4) emerge naturally, resulting in a model that is rich in structure.

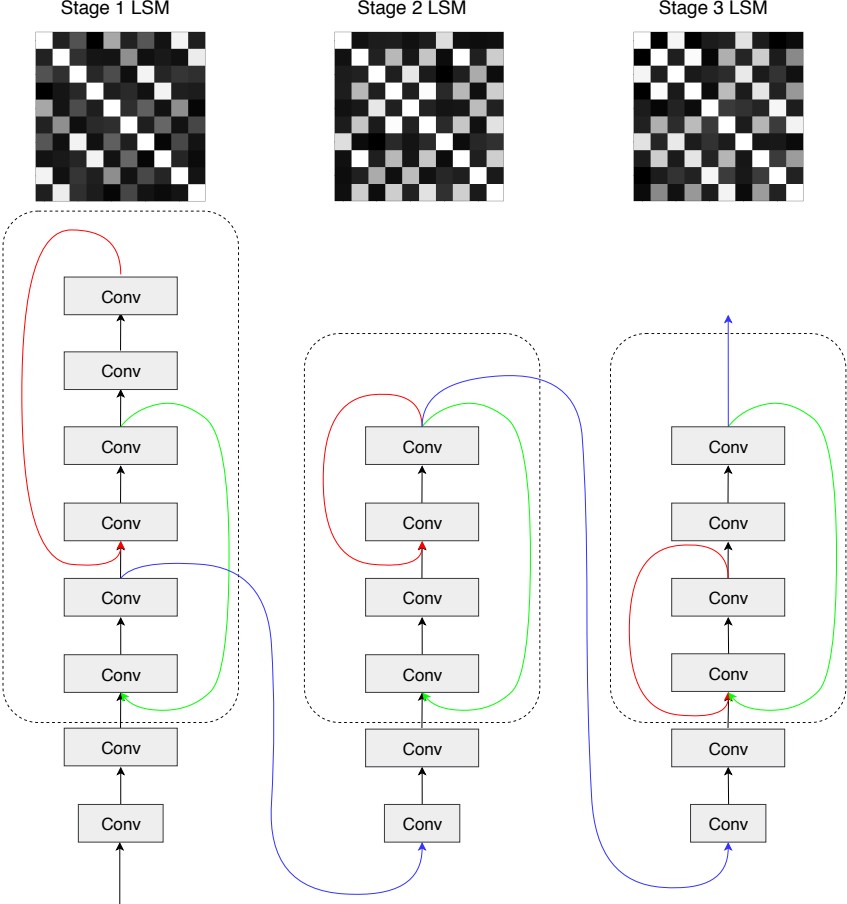

Figure 6: SWRN 40-8-8 (8 parameter templates shared among groups of $\frac{40-4}{3} - 2 = 10$ layers) trained with soft parameter sharing on CIFAR-10. Each stage (originally with 12 layers – the first two do not participate in parameter sharing) can be folded to yield blocks with complex recurrences. For clarity, we use colors to indicate the computational flow: red takes precedence over green, which in turn has precedence over blue. Colored paths are only taken once per stage. Although not trivial to see, recurrences in each stage's folded form are determined by row/column repetitions in the respective Layer Similarity Matrix. For example, for stage 2 we have $S_{5,3} \approx S_{6,4} \approx 1$, meaning that layers 3, 4, 5 and 6 can be folded into layers 3 and 4 with a loop (captured by the red edge). The same holds for $S_{7,1}$, $S_{8,2}$, $S_{9,3}$ and $S_{10,4}$, hence after the loop with layers 3 and 4, the flow returns to layer 1 and goes all the way to layer 4, which generates the stage's output. Even though there is an approximation when folding the network (in this example, we are tying layers with similarity close to 0.8), the impact on the test error is less than 0.3%. Also note that the folded model has a total of 24 layers (20 in the stage diagrams, plus 4 which are not shown, corresponding to the first layer of the network and three $1 \times 1$ convolutions in skip-connections), instead of the original 40.

Figure 7: LSMs of a SWRN 40-8-8 (composed of 3 stages, each with 10 layers sharing 8 templates) trained on CIFAR-10 for 5 runs with different random seeds. Although the LSMs differ across different runs, hard parameter sharing can be observed in all cases (off-diagonal elements close to 1, depicted by white), characterizing implicit recurrences which would enable network folding. Moreover, the underlying structure is similar across runs, with hard sharing typically happening among layers $i$ and $i + 2$, leading to a "chessboard" pattern.

Figure 8: LSMs of a SWRN 40-8-8 (composed of 3 stages, each with 10 layers sharing 8 templates) at different epochs during training on CIFAR-10. The transition from an identity matrix to the final LSM happens mostly in the beginning of training: at epoch 50, the LSM is almost indistinguishable from the final LSM at epoch 200, and most of the final patterns are observable already at epoch 25.

## B    INITIALIZATION OF COEFFICIENTS

During our initial experiments, we explored different initializations for the coefficients $\boldsymbol{\alpha}$ of each layer, and observed that using an orthogonal initialization (Saxe et al., 2013) resulted in superior performance compared to uniform or normal initialization schemes.

Denote $\boldsymbol{A}$ as the $L \times k$ matrix ($L$ is the number of layers sharing parameters and $k$ the number of templates) with each $i$'th row containing the coefficient of the $i$'th layer $\boldsymbol{\alpha}^{(i)}$. We initialize it such that $\boldsymbol{A}^T \boldsymbol{A} = \boldsymbol{I}$, leading to $\forall_i, \langle \boldsymbol{\alpha}^{(i)}, \boldsymbol{\alpha}^{(i)} \rangle = 1$ and $\forall_{i \neq j}, \langle \boldsymbol{\alpha}^{(i)}, \boldsymbol{\alpha}^{(j)} \rangle = 0$. While our choice for this is mostly empirical, we believe that there is likely a connection with the motivation for using orthogonal initialization for RNNs.

Moreover, we discovered that other initialization options for $\boldsymbol{A}$ work similarly to the orthogonal one. More specifically, either initializing $\boldsymbol{A}$ with the identity matrix when $L = k$ (which naturally leads to $\boldsymbol{A}^T \boldsymbol{A} = \boldsymbol{I}$) or enforcing some sparsity (initialize $\boldsymbol{A}$ with a uniform or normal distribution and randomly setting half of its entries to zero) performs similarly to the orthogonal initialization in a consistent manner. We believe the sparse initialization to be the simplest one, as each coefficient $\boldsymbol{\alpha}$ can be initialized independently.

Finally, note that having $\boldsymbol{A}^T \boldsymbol{A} = \boldsymbol{I}$ results in the Layer Similarity Matrix also being the identity at initialization (check that $S_{i,j} = \frac{|\langle \boldsymbol{\alpha}^{(i)}, \boldsymbol{\alpha}^{(j)} \rangle|}{\|\boldsymbol{\alpha}^{(i)}\| \|\boldsymbol{\alpha}^{(j)}\|} = \frac{|(\boldsymbol{A}^T \boldsymbol{A})_{i,j}|}{\|\boldsymbol{\alpha}^{(i)}\| \|\boldsymbol{\alpha}^{(j)}\|}$, so if $(\boldsymbol{A}^T \boldsymbol{A})_{i,j} = 1$, then $S_{i,j} = 1$, and the same holds for 0. Surprisingly, even though the orthogonal initialization leads to a LSM that has no structure in the beginning of training, the rich patterns that we observe still emerge naturally after optimization.

