# OpenReview forum: "Learning Implicitly Recurrent CNNs Through Parameter Sharing"
_ICLR.cc/2019/Conference_

### Official Review · AnonReviewer2 · 2018-11-02
**Interesting idea of using parameter sharing scheme to explore network structure; experiments can be stronger**

**Rating:** 6
**Confidence:** 4

**Review:**

Authors propose a parameter sharing scheme by allowing parameters to be reused across layers. It further makes connection between traditional CNNs with RNNs by adding additional regularization and using hard sharing scheme.

The way of parameter sharing is similar to the filter prediction method proposed in Rebuff et al’s work, where they model a convolutional layer’s parameters as a linear combination of a bank of filters and use that to address difference among multiple domains.

Sylvestre-Alvise Rebuffi, Hakan Bilen, Andrea Vedaldi, Learning multiple visual domains with residual adapters, NIPS 2017.

The discussion on the connection between coefficients for different layers and a network’s structure and visualization of layer similarity matrix is interesting. Additional regularization can further encourage a recurrent neural network to be learned.

However, they only experiment with one or two templates and advantage on accuracy and model size  over other methods is not very clear.

---

> ### Author Response · Authors · 2018-11-22
> **We thank you for the feedback, and address specific points below.**
>
> 1 - The way of parameter sharing is similar to the filter prediction method proposed in Rebuff et al’s work, where they model a convolutional layer’s parameters as a linear combination of a bank of filters and use that to address difference among multiple domains.
>
> Thank you for pointing out the work of Rebuffi et al.  There are similarities in some technical aspects, as both our work and theirs involve learning a bank of filters and mixing coefficients.  However, the overall goal of our work is entirely different from that of Rebuffi et al., as are the additional technical tools (e.g. layer similarity matrix, network folding) that we develop.
>
> Rebuffi et al. focus on domain-adaptability and transfer-learning, while we focus on parameter reduction and architecture discovery. Because of this, we believe the findings of both papers (ours and Rebuffi et al.) are extremely complementary, as together they show the potential of ‘template learning’ both for the multi-domain setting (as it offers better domain-adaptation) and for single-domain (as it offers better performance, parameter reduction, and potentially simpler models). Note that we introduce novel tools for manipulating single-domain networks, yielding the ability to fold them into recurrent forms, that have no parallel in Rebuffi et al. We will add a discussion of the work of Rebuffi et al. in the final version of our manuscript.
>
> 2 - However, they only experiment with one or two templates and advantage on accuracy and model size over other methods is not very clear.
>
> Actually, our experiments include models with the number of templates ranging from 1 to 20 (per sharing group). More specifically, the CIFAR experiments (refer to Tables 1, 2 and Figure 3) consist of models with between 1 and 6 templates: the SWRN-L-w-k models where k is omitted (e.g. SWRN 28-10) have one template per sharing layer, meaning k=6 for 28-layered models.  As we believe that omitting k to represent one template per layer can lead to confusion, we will revise this notation in a forthcoming update of the paper,
>
> For these same experiments, we focused on the regime with few templates since our goal is to decrease parameter redundancy: we believe one of our most important findings is that networks with k=1 or k=2 manage to outperform the same models with larger k, as the latter have significantly more capacity.
>
> In our last experiments (in Section 4.4), the two SCNN models have a total of 20 templates shared among 20 convolutional layers. In this setting, the SCNN significantly outperforms the CNN, and adapts faster to curriculum changes (Figure 5b, compare blue and orange curves).
>
> As for advantage on accuracy and model size, Table 1 shows that we can achieve both performance increase and parameter reduction: the SWRN 28-18-2 model not only outperforms SWRN 28-18, but also has less than half of its parameters. It also outperforms the ResNeXt model, which, even though it has bottleneck layers, still has more parameters than SWRN 28-18-2. We would also like to point out that all results are average of 5 runs (except the curves in Figure 5b).

---

### Official Review · AnonReviewer1 · 2018-11-02
**a promising proposal that exploits the over-parameterization nature of neural nets to reduce the model size**

**Rating:** 7
**Confidence:** 3

**Review:**

This work is motivated by the widely recognized issue of over-parameterization in modern neural nets, and proposes a clever template sharing design to reduce the model size. The design is sound, and the experiments are valid and thorough. The writing is clear and fluent.

The reviewer is not entirely sure of the originality of this work. According to the sparse 'related work' section, the contribution is novel, but I will leave it to the consensus of others who are more versed in this regard.

The part that I find most interesting is the fact that template sharing helps with the optimization without even reducing the number of parameters, as illustrated in CIFAR from Table 1. The trade-off of accuracy and parameter-efficiency is overall well-studied in CIFAR and ImageNet, although results on ImageNet is not as impressive.

Regarding the coefficient alpha, I'm not sure how cosine similarity is computed. I have the impression that each layer has its own alpha, which is a scalar. How is cosine similarity computed on scalars?

In the experiments, there's no mentioning of the regularization terms for alpha, which makes me think it is perhaps not important? What is the generic setup?

In summary, I find this work interesting, and with sufficient experiments to backup its claim. On the other hand, I'm not entirely sure of its novelty/originality, leaving this part open to others.

---

> ### Author Response · Authors · 2018-11-22
> **Thank you for the feedback and specific questions, which we address in detail below.**
>
> With respect to novelty, we do not believe there is any existing work that utilizes a parameter sharing scheme toward the objective we accomplish here: training a deep network and then folding it into a recurrent form.  Please also see our detailed reply to AnonReviewer2.
>
> 1- Regarding the coefficient alpha, I'm not sure how cosine similarity is computed. I have the impression that each layer has its own alpha, which is a scalar. How is cosine similarity computed on scalars?
>
> Each layer i has its own alpha parameter, denoted by alpha^(i) in the manuscript (refer to equation 1 on page 3), but each alpha is a k-dimensional vector, where k is the number of templates to which that layer has access. For the SWRN-L-w-k models we use in the experiments, the same k denotes the number of templates each layer can use, so the dimensionality of each alpha ranges from 1 (in this case it’s just a scalar) to 6 in our experiments.
>
> We made alpha bold in the current version of the manuscript to clarify that it is a vector (except when k=1).
>
> 2 - In the experiments, there's no mentioning of the regularization terms for alpha, which makes me think it is perhaps not important? What is the generic setup?
>
> We tried applying L2 regularization to the alpha parameters in our initial experiments, but observed a performance drop, so all reported experiments have no regularization on the alphas.
>
> As for the recurrence regularizer described in the end of Section 3.2, where we regularize the Layer Similarity Matrix, it was only used for the experiments in Section 4.4 -- more specifically, the “SCNN, lambda_R = 0.01” model depicted in Figure 5. It was not used for any other experiments, meaning that the observed patterns (e.g. the Layer Similarity Matrices in Figure 4) emerge naturally during optimization, where neither the alphas nor the LSMs had any regularization.

---

### Official Review · AnonReviewer3 · 2018-11-05
**Interesting approach to weight sharing among CNN layers via shared weight templates, well written, convincing results.**

**Rating:** 8
**Confidence:** 4

**Review:**

The manuscript introduces a novel and interesting approach to weight sharing among CNNs layers, by learning linear combinations of shared weight templates. This allows parameter reduction, better sample efficiency. Furthermore, the authors propose a very simple way to inspect which layers choose similar combinations of template, as well as to push the network toward using similar combinations at each layer. This regularization term has a clear potential for computation reuse on dedicated hardware. The paper is well written, the method is interesting, the results are convincing and thoroughly conducted. I recommend acceptance.

1) It would be interesting to explore how often the layer parameters converge to similar weights and how similar. To this end I suggest to plot a 2d heatmap representing the similarity matrices between every pair of layers.

2) Figure 1 is not of immediate interpretability. Especially for the middle figure, what does the dotted box represent? What is the difference between weights and templates? Also it’s unclear which of the three options corresponds to the proposed method. I would have thought the middle one, but the text seems to indicate it is the rightmost one instead.

3) How are the alphas initialized? How fast are their transitions? Do they change smoothly over training? Do they evolve rapidly and plateau to a fixed value or keep changing during training? It would be really interesting to plot their value and discuss their evolution.

4) While the number of learned parameters is indeed reduced when the templates are shared among layers - which could lead to better sample efficiency - I am not sure whether the memory footprint on GPU would change (i.e., I believe that current frameworks would allocate the same kernel n-times if the same template was shared by n layers, but I am not certain). Although the potential reduction of the number of trainable parameters is an important result by itself, I wonder if what you propose would also allow to run bigger models on the same device or not, without heavy modifications of the inner machineries of pyTorch or Tensorflow.  Can you comment on this? Also note that the soft sharing regularization scheme that you propose can be of great interest for FPGA hardware implementations, that benefit a lot from module reuse. You could mention that in the paper.

5) Sec 4.1, the number of layers in one group is defined as (L-4)/3. It’s unclear to me where the 4 comes from. Also on page 6, k = (L-2)/3 - 2 is said to set one template per layer. I thought the two formulas would be the same in that case. What am I missing? Is it possible that one of the two formulas contain a typo (I believe that at the very least it should be either (L-2) or (L-4) in both cases)?

6) Sec 4.1, I find the notation SWRN-L-w-k and SWRN-L-w confusing. My suggestion is to set k to be the total number of templates (as opposed to the number of templates *per group of layers*), which makes it much easier to relate to, and most importantly allows for an immediate comparison with the capacity of the vanilla model. As a side effect, it also makes it very easy to spot the configuration with one template per layer (SWRN-L-w-L) thus eliminating the need for an ad-hoc notation to distinguish it.

7) The authors inspect how similar the template combination weights alphas are among layers. It would also be interesting to look into what is learned in the templates. CNN layers are known to learn peculiar and somewhat interpretable template matching filters. It would be really interesting to compare the filters learned by a vanilla network and its template sharing alternative. Also, I would welcome an analysis of which templates gets chosen the most at each layer in the hierarchy. It would be compelling if some kind of pattern of reuse emerged from learning.

8) Sec 4.4, it is unclear to me what can be the contribution of the 1x1 initial convolution, since it will see no context and all the information at the pixel level can be represented by a binary bit. Also, are the 3x3 convolutions “same” convolutions? If not, how are the last feature maps upscaled to be able to predict at the original resolution?

9) At the end of sec 4.4 the authors claim that the SCNN is “also advantaged over a more RNN-like model”. I fail to understand how to process this sentence, but I have a feeling that it’s incorrect to make any claims to the performance of “RNN-like models” as such a model was not used as a baseline in the experiments in any way. Similarly, in the conclusions I find it a bit stretched to claim that you can “gain a more flexible form of behavior typically attributed to RNNs”. While it’s true that the proposed network can in theory learn to reuse the same combination of templates, which can be mapped to a network with recursion, the results in this direction don’t seem strong enough to draw any conclusion and a more in-depth comparison against RNN performance would be in order before making any claim in this direction.


MINOR
- Sec3: I wouldn’t say the parameters are shared among layers in LSTMs, but rather among time unrolls.
- One drawback of the proposed method is that the layers are constrained to have the same shape. This is not a major disadvantage, but is still a constraint that would be good to make more explicit in the description of the model.
- Sec3, end of page 3: does the network reach the same accuracy as the vanilla model when k=L? Also, does the network use all the templates? How is the distribution of the alpha weights across layers in this case?
- Sec3.1, the V notation makes the narrative unnecessarily heavy. I suggest to drop it and refer directly to the templates T. Also the second part of the section, with examples of templates, doesn’t add much in my opinion and would be better depicted with a figure.
- Sec3.1, the e^(i) notation can be confused with an exp. I suggest to replace it with the much more common 1_{i=j}.
- Figure 2 depicts the relation between the LSM matrix and the topology of the network. This should be declared more clearly in the caption, in place of the ambiguous “capturing implicit recurrencies”. Also, the caption should explain what black/white stand for as well, and possibly quickly describe what the LSM matrix is. Also, it would be more clear that the network in the middle is equivalent to that on the right if the two were somehow connected in the figure. To this end they could, e.g., share a single LSM matrix among them. Finally, if possible try and put the LSM matrices on top of the related network, so that it’s clear which network they refer to. Sec 3.2 should also refer to Fig2 I believe.
- Table 1: I suggest to leave the comment on the results out of the caption, since it’s already in the main text.
- Table 2: rather than using blue, I suggest to underline the overall best results, so that it’s visible even if the paper is printed in B&W.
- Fig 3, I would specify that it’s better viewed in color
- Discussion: I feel the discussion of Table 1 is a bit difficult to follow. It could be made easier by reporting the difference in test error against the corresponding vanilla model (e.g., “improves the error rate on CIFAR10 by 0.26%”, rather than reporting the performance of both models)
- Fig 4, are all the stages the same and is the network in the left one such stages? If so, update the caption to make it clear please.
- Fig 4, which lambda has been used? Is it the same for all stages?
- Fig 5, specify that the one on the right is the target grid. Also, I believe that merging the two figures would make it easier to understand (e.g., some of the structure in the target comes from how the obstacles are placed, which requires to move back and forth from input to target several times to understand)
- Sec 4.4, space permitting, I would like to see at least one sample of what kind of shortest path prediction the network can come up with.



A few typos:
    * End of 3.2: the closer elements -> the closer the elements
    * Parameter efficiency: the period before re-parametrizing should probably be a comma?
    * Fig 4, illustration of stages -> illustration of the stages
    * End of pag7, an syntetic -> a syntetic

---

> ### Author Response · Authors · 2018-11-22
> **[4/4] Thank you.  We appreciate the detailed feedback and questions.**
>
> MINOR
> - Sec3: I wouldn’t say the parameters are shared among layers in LSTMs, but rather among time unrolls.
>
> We changed to “shared among all time steps” in the manuscript.
>
> - One drawback of the proposed method is that the layers are constrained to have the same shape. This is not a major disadvantage, but is still a constraint that would be good to make more explicit in the description of the model.
>
> We added an explicit mention of this constraint in Section 3 (paragraph following Equation 1).
>
> - Sec3, end of page 3: does the network reach the same accuracy as the vanilla model when k=L? Also, does the network use all the templates? How is the distribution of the alpha weights across layers in this case?
>
> Yes, in all our experiments having one template per layer resulted in better accuracy than the vanilla model (for example, refer to Table 1, models WRN 28-10 and SWRN 28-10). In all our experiments, each template is used by at least one layer, however some layers do not use all templates (having one component of the coefficient vector alpha very close to zero). As mentioned previously, we will add plots of the coefficients to the final version.
>
> - Sec3.1, the V notation makes the narrative unnecessarily heavy. I suggest to drop it and refer directly to the templates T. Also the second part of the section, with examples of templates, doesn’t add much in my opinion and would be better depicted with a figure.
> - Sec3.1, the e^(i) notation can be confused with an exp. I suggest to replace it with the much more common 1_{i=j}.
>
> We removed the V notation and refer to templates directly. We have also removed the first example (\alpha^(i) = e^(i)) for simplicity (and as the second one is more relevant for our work).
>
> - Figure 2 depicts the relation between the LSM matrix and the topology of the network. This should be declared more clearly in the caption, in place of the ambiguous “capturing implicit recurrencies”. Also, the caption should explain what black/white stand for as well, and possibly quickly describe what the LSM matrix is. Also, it would be more clear that the network in the middle is equivalent to that on the right if the two were somehow connected in the figure. To this end they could, e.g., share a single LSM matrix among them. Finally, if possible try and put the LSM matrices on top of the related network, so that it’s clear which network they refer to. Sec 3.2 should also refer to Fig2 I believe.
>
> We updated Figure 2 and its caption (due to space constraints we could not place the LSM matrices on top of the networks, so we added dashed lines to make it clear which network each LSM matrix corresponds to).
>
> - Table 1: I suggest to leave the comment on the results out of the caption, since it’s already in the main text.
>
> We shortened the discussion on the results both in the caption of Table 1 and in Section 4.1.
>
> - Table 2: rather than using blue, I suggest to underline the overall best results, so that it’s visible even if the paper is printed in B&W.
>
> We changed from blue to underline to indicate best results.
>
> - Fig 3, I would specify that it’s better viewed in color
>
> Added to the caption.
>
> - Discussion: I feel the discussion of Table 1 is a bit difficult to follow. It could be made easier by reporting the difference in test error against the corresponding vanilla model (e.g., “improves the error rate on CIFAR10 by 0.26%”, rather than reporting the performance of both models)
>
> We updated the manuscript: instead of reporting the errors of both models, we report the relative error decrease (we believe the absolute error decrease might not be meaningful since the scale of the errors is no longer reported in the discussion).
>
> - Fig 4, are all the stages the same and is the network in the left one such stages? If so, update the caption to make it clear please.
>
> The diagram on the left represents the architecture of each stage of the network, and all 3 stages have the same topology. We updated the caption to mention this explicitly.
>
> - Fig 4, which lambda has been used? Is it the same for all stages?
>
> The recurrence regularizer has not been applied to any experiment except for the last one (Section 4.4): the patterns observed in the LSM, which enabled folding, have emerged naturally during training.
>
> - Fig 5, specify that the one on the right is the target grid. Also, I believe that merging the two figures would make it easier to understand (e.g., some of the structure in the target comes from how the obstacles are placed, which requires to move back and forth from input to target several times to understand)
>
> We have merged the two figures together.
>
> - Sec 4.4, space permitting, I would like to see at least one sample of what kind of shortest path prediction the network can come up with.
>
> We will aim to add at least one example in the final version of the manuscript.

---

> > ### Comment · AnonReviewer3 · 2018-11-22
> > **Thank you for the detailed and satisfactory reply.**
> >
> > Thank you for addressing all my comments in detail, your replies satisfactorily cover all the issues I raised. I look forward to reading the final version of the manuscript.

---

> ### Author Response · Authors · 2018-11-22
> **[3/4] Thank you. We appreciate the detailed feedback and questions.**
>
> 8 - Sec 4.4, it is unclear to me what can be the contribution of the 1x1 initial convolution, since it will see no context and all the information at the pixel level can be represented by a binary bit. Also, are the 3x3 convolutions “same” convolutions? If not, how are the last feature maps upscaled to be able to predict at the original resolution?
>
> We wanted all 3x3 convolutions to participate in parameter sharing, and to do so they need to have the same number of input and output channels. Without the initial 1x1 convolution, the next convolution (the first 3x3 one) would have 2 input channels instead of 32, like all other 3x3 convolutions. The 1x1 kernel size was chosen since the only purpose of this convolution is to change the number of channels.
>
> All 3x3 convolutions are “same”, with outputs having the same spatial resolution as the input.
>
>
> 9 - At the end of sec 4.4 the authors claim that the SCNN is “also advantaged over a more RNN-like model”. I fail to understand how to process this sentence, but I have a feeling that it’s incorrect to make any claims to the performance of “RNN-like models” as such a model was not used as a baseline in the experiments in any way. Similarly, in the conclusions I find it a bit stretched to claim that you can “gain a more flexible form of behavior typically attributed to RNNs”. While it’s true that the proposed network can in theory learn to reuse the same combination of templates, which can be mapped to a network with recursion, the results in this direction don’t seem strong enough to draw any conclusion and a more in-depth comparison against RNN performance would be in order before making any claim in this direction.
>
> We will rephrase our claims to make them more precise. By ‘RNN-like model’ we meant the SCNN with lambda_R = 0.01 (recurrence regularizer), where all 20 layers become maximally similar in the first 10 epochs (all elements in the LSM get very close to 1), meaning that the filters used by all 20 convolutions are (roughly) the same (up to scaling). While in practice this can be seen as a RNN, we agree that our wording should be more specific. We added this observation (on the LSM of the SCNN with the recurrence regularizer) to the manuscript.

---

> ### Author Response · Authors · 2018-11-22
> **[2/4] Thank you. We appreciate the detailed feedback and questions.**
>
> 4 - While the number of learned parameters is indeed reduced when the templates are shared among layers - which could lead to better sample efficiency - I am not sure whether the memory footprint on GPU would change (i.e., I believe that current frameworks would allocate the same kernel n-times if the same template was shared by n layers, but I am not certain). Although the potential reduction of the number of trainable parameters is an important result by itself, I wonder if what you propose would also allow to run bigger models on the same device or not, without heavy modifications of the inner machineries of pyTorch or Tensorflow. Can you comment on this? Also note that the soft sharing regularization scheme that you propose can be of great interest for FPGA hardware implementations, that benefit a lot from module reuse. You could mention that in the paper.
>
> Thanks for pointing out the potential for FPGA implementations.
> There are at least two settings where our method can lead to infrastructural advantages:
>
> (i) During training PyTorch and Tensorflow both store the generated weights for each layer (to compute gradients in the backward pass), but at test time this is not the case: once the weights have been generated and used for the convolution, the memory is freed, and the only parameters that have to be stored are the alphas and the templates (which are only allocated once, regardless of how many layers use them).  Returning to the question of training, we believe that dedicated implementations could lead to significant memory savings.  For example, instead of storing the weights generated for each layer, training could recompute them in the backward pass (such recomputation is a cheap operation).
>
> (ii) In the distributed training setting with multiple machines, gradients have to be communicated at each optimization round. In the case where our method offers roughly 50% parameter reduction (Tables 1 and 2), communication cost would also be decreased by 50%.
>
>
> 5 - Sec 4.1, the number of layers in one group is defined as (L-4)/3. It’s unclear to me where the 4 comes from. Also on page 6, k = (L-2)/3 - 2 is said to set one template per layer. I thought the two formulas would be the same in that case. What am I missing? Is it possible that one of the two formulas contain a typo (I believe that at the very least it should be either (L-2) or (L-4) in both cases)?
>
> The 4 is specific for Wide ResNets, where we have a first convolution with 3 input channels (RBG of the input image) and 3 convolutions with 1x1 kernels in skip-connections (one for every first skip-connection of each stage). These layers do not participate in sharing since no other layers in the network have the same parameter shapes.
> The k = (L-2)/3 - 2 from page 6 is indeed a typo which we fixed -- thanks for pointing it out.
>
>
> 6 - Sec 4.1, I find the notation SWRN-L-w-k and SWRN-L-w confusing. My suggestion is to set k to be the total number of templates (as opposed to the number of templates *per group of layers*), which makes it much easier to relate to, and most importantly allows for an immediate comparison with the capacity of the vanilla model. As a side effect, it also makes it very easy to spot the configuration with one template per layer (SWRN-L-w-L) thus eliminating the need for an ad-hoc notation to distinguish it.
>
> We agree with your suggestion and will revise the notation in a forthcoming update of the paper. However, note that the configuration with one template per layer will not be denoted by SWRN-L-w-L since some layers do not participate in parameter sharing (see above).
>
>
> 7 - The authors inspect how similar the template combination weights alphas are among layers. It would also be interesting to look into what is learned in the templates. CNN layers are known to learn peculiar and somewhat interpretable template matching filters. It would be really interesting to compare the filters learned by a vanilla network and its template sharing alternative. Also, I would welcome an analysis of which templates gets chosen the most at each layer in the hierarchy. It would be compelling if some kind of pattern of reuse emerged from learning.
>
> We agree that inspecting the learned behavior of our shared networks and contrasting with standard CNNs in a worthwhile effort.  Given that visualizing and understanding CNNs is itself an area of active research, performing a thorough analysis in this respect is probably beyond the scope of the current paper; it would require substantial additional experiments.
>
> As for which templates get chosen the most, we have plots of the alphas themselves, which indicate which templates are most important to each layer. We will add them to the final version.

---

> ### Author Response · Authors · 2018-11-22
> **[1/4] Thank you. We appreciate the detailed feedback and questions.**
>
> 1 - It would be interesting to explore how often the layer parameters converge to similar weights and how similar. To this end I suggest to plot a 2d heatmap representing the similarity matrices between every pair of layers.
>
> Thanks for the suggestion; we agree that this is a worthwhile experiment. The LSM matrices (Figure 4 & 6) already capture the similarity between every pair of layer for each stage: S_i,j is the absolute cosine similarity between the alpha parameters of layers i and j, so S_i,j ~ 1 (white) indicate that layers i and j have learned similar alphas (up to scaling).  However, exploring how often this alignment occurs is definitely relevant for the work. We will aim to include such experimental results in the final version of the paper.
>
>
> 2 -  Figure 1 is not of immediate interpretability. Especially for the middle figure, what does the dotted box represent? What is the difference between weights and templates? Also it’s unclear which of the three options corresponds to the proposed method. I would have thought the middle one, but the text seems to indicate it is the rightmost one instead.
>
> The middle figure attempts to represent the case where filters W are generated from coefficients alpha and templates T through some function W = f(alpha, T). The dotted box represents that, unlike in the left figure (illustrating a network without parameter sharing), the weights are no longer parameters, but are generated from the coefficients alpha and the templates.  The distinction between weights and templates is that each has a different purpose: the weights are used as filters for the convolutional layers, while templates are used to generate weights (and are shared among many layers).
>
> Our method is illustrated by both the middle and the right figures: for the mapping W = f(alpha, T), whenever f(.) is a linear function, the middle and right figures become equivalent due to associativity (explained in more detail in section 3.1). They offer two perspectives for our technique.  We added a more detailed explanation to the caption of Figure 1. We updated the caption to make this more explicit.
>
>
> 3 - How are the alphas initialized? How fast are their transitions? Do they change smoothly over training? Do they evolve rapidly and plateau to a fixed value or keep changing during training? It would be really interesting to plot their value and discuss their evolution.
>
> In our ablation experiments, we observed that typical initializations (normal, uniform) are not good choices for the alphas. On the other hand, sparse and orthogonal initializations work better; we used orthogonal in the reported experiments. We added a discussion on initialization to the appendix.
>
> The transitions are fast and happen mostly in the first 20 epochs of training, forming the patterns present in the Layer Similarity Matrices. After that point, typically similar layers become more similar and dissimilar layers become less similar, in a smooth fashion. We will add an example of evolution of a LSM during training in the appendix for the final version.

---

### Meta-Review · Area_Chair1 · 2018-12-17
**a promising idea**

**Confidence:** 5
**Recommendation:** Accept (Poster)

**Metareview:**

This paper proposed an interesting approach to weight sharing among CNN layers via shared weight templates to save parameters. It's well written with convincing results. Reviewers have a consensus on accept.